# Construction of Multiple Guide RNAs in CRISPR/Cas9 Vector Using Stepwise or Simultaneous Golden Gate Cloning: Case Study for Targeting the *FAD2* and *FATB* Multigene in Soybean

**DOI:** 10.3390/plants10112542

**Published:** 2021-11-22

**Authors:** Won-Nyeong Kim, Hye-Jeong Kim, Young-Soo Chung, Hyun-Uk Kim

**Affiliations:** 1Department of Bioindustry and Bioresource Engineering, Plant Engineering Research Institute, Sejong University, Seoul 05006, Korea; k9312@sju.ac.kr; 2Department of Molecular Genetics, College of Natural Resources and Life Science, Dong-A University, Busan 49315, Korea; hjkim83@dau.ac.kr (H.-J.K.); chungys@dau.ac.kr (Y.-S.C.)

**Keywords:** CRISPR/Cas9, multiple sgRNA, polycistronic-tRNA-gRNA, Golden Gate cloning, type IIS restriction enzyme, *FAD2*, *FATB*, soybean

## Abstract

CRISPR/Cas9 is a commonly used technique in reverse-genetics research to knock out a gene of interest. However, when targeting a multigene family or multiple genes, it is necessary to construct a vector with multiple single guide RNAs (sgRNAs) that can navigate the Cas9 protein to the target site. In this protocol, the Golden Gate cloning method was used to generate multiple sgRNAs in the Cas9 vector. The vectors used were pHEE401E_UBQ_Bar and pBAtC_tRNA, which employ a one-promoter/one-sgRNA and a polycistronic-tRNA-gRNA strategy, respectively. Golden Gate cloning was performed with type IIS restriction enzymes to generate gRNA polymers for vector inserts. Four sgRNAs containing the pHEE401E_UBQ_Bar vector and four to six sgRNAs containing the pBAtC_tRNA vector were constructed. In practice, we constructed multiple sgRNAs targeting multiple genes of *FAD2* and *FATB* in soybean using this protocol. These three vectors were transformed into soybeans using the *Agrobacterium-*mediated method. Using deep sequencing, we confirmed that the T0 generation transgenic soybean was edited at various indel ratios in the predicted target regions of the *FAD2* and *FATB* multigenes. This protocol is a specific guide that allows researchers to easily follow the cloning of multiple sgRNAs into commonly used CRISPR/Cas9 vectors for plants.

## 1. Introduction

Since gene-knockout technology has become available in plant research, reverse genetics is now widely used to establish the logic for proving the expected hypothesis. Reverse genetics is used to elucidate hypotheses by knocking out or modifying a gene of interest (GOI) and examining the altered phenotype of a model species [1]. It offers fewer samples than forward genetics but shows a greater association between genes and phenotypes. Advances in biotechnology have led to the development of many methods for manipulating GOIs, of which the most frequently used are zinc finger nucleases (ZFNs) and transcription activator-like effector nucleases (TALENs) before the advent of clustered regulatory interspaced short palindromic repeats (CRISPR) systems [1]. All these programmable nucleases can cause double-strand breaks (DSB) and induce DNA repair systems such as homologous recombination (HR) or non-homologous end-joining (NHEJ), which fundamentally reform or delete original base sequences. Among such programmable nucleases, the CRISPR system, which uses Cas9 protein, is currently the most feasible and commonly used method in research. To design an experiment using the CRISPR/Cas9 system, the most important step is to correctly set a single guide RNA (sgRNA). The sgRNA is a 20-nucleotide long, single-stranded RNA that combines with the Cas9 protein and provides information about the target site through a complementary sequence.

Several methods for constructing multiple sgRNA vectors have been reported to simultaneously knock out multiple genes [2]. The Golden Gate assembly is a common approach for constructing a multi-sgRNA vector [3,4,5]. The procedure starts with generating insert fragments that are ligated into the subcloning vector. The insert can be generated by the template vector PCR or oligonucleotide annealing. The next step is subcloning, which makes sgRNA fragments available for Golden Gate assembly. Finally, after the DNA sequence verification of the subcloning vector, the Cas9 vector and subcloned vectors are mixed in one pot for the Golden Gate assembly. However, the above-mentioned methods require a special pair of subcloning and CRISPR/Cas9 vectors for Golden Gate assembly. Therefore, a more ubiquitous method is required for cloning multiple sgRNAs using type IIS restriction enzymes, especially in laboratories that only have a general CRISPR/Cas9 vector.

The protocol in this study describes the construction of a multi-sgRNA vector that contains at least four sgRNAs, using the common CRISPR/Cas9 vector based on type IIS restriction enzyme ligation. This protocol does not require a subcloning step, which has the advantage of reducing the required time by approximately 2–3 days. Here, two strategies are described to construct two different vectors using either one promoter per sgRNA or polycistronic tRNA-gRNA. The pHEE401E vector has a type IIS restriction-enzyme-recognition site for ligation [6]. The other is a pBAtC-tRNA vector that uses a polycistronic tRNA-gRNA expression system and has two *AarI* type IIS restriction enzyme sites. Originally, these two vectors could carry two sgRNAs. However, using a modified primer set and ligation strategy, the pHEE401E vector can carry four sgRNAs, and the pBAtC-tRNA vector can carry four to six sgRNAs.

## 2. Experimental Design

Two main vectors were used in this procedure. The original vectors were pHEE401E [7] and pBAtC [8], which can be obtained from local researchers or through non-profit organizations such as Addgene (https://www.addgene.org, accessed on 21 November 2021). Each insert in the pHEE401E vector consists of a scaffold, a terminator for upstream sgRNA2, a promoter for its own sgRNA3, and sgRNA3, except for the topmost insert that starts with sgRNA1 (Figure 1). This pattern also applies to sgRNA3 and sgRNA4. The sites for ligation have identical sticky ends between inserts after digestion, indicating that stepwise ligation is important for acquiring the intended insert polymer.

The pBAtC vector has simpler inserts than the pHEE401E vector. There are only scaffolds and tRNAs between the two sgRNAs. In addition, the ligation site in the middle of the sgRNA can generate different sticky ends after digestion, which means that simultaneous ligation of the ordered insert polymer is possible.

### 2.1. pHEE401E_ UBQ_Bar_4sgRNA Cloning

#### 2.1.1. Materials

The pHEE401E_UBQ_Bar vector is a version in which the egg-specific promoter for Cas9 expression in the pHEE401E vector was replaced with the *UBQ10* promoter, and the hygromycin resistance gene was replaced with the *bar* gene for plant transgenic selection (Appendix A) (this vector was kindly provided by Mr. Mid-Eum Park, Department of Molecular Biology, Sejong University, Seoul, Korea).pCBC_DT1T2 (Appendix A) (kindly provided by Professor Mi Chung Suh, Department of Life Science, Sogang University, Seoul, Korea).KOD-plus-(TOYOBO, Osaka, Japan; Cat. No. KOD-201 200U).The QIAquick PCR Purification Kit (Cat. No. 28104).The QIAquick Gel Extraction Kit (Cat. No. 28704).T4 DNA ligase (TaKaRa, Kusatsu, Japan; Cat. No. 2011A).10X T4 DNA ligase buffer (TaKaRa, Japan; Cat. No. 2011A).*Aar*I restriction enzyme (Thermo Fisher, Waltham, MA, USA; Cat. No. ER1581).*Bbs*I restriction enzyme (NEB, Ipswich, MA, USA; Cat. No. R0539S).*BsaI*-HFv2 restriction enzyme (NEB, Ipswich, MA, USA.; Cat. No. R3733S).

#### 2.1.2. Equipment

TaKaRa PCR Thermal Cycler Dice^®^ Touch (TaKaRa, Japan; Cat. No. TP350).Mupid-exU (Advanced, Nomi, Japan; Cat. No. AD140).

#### 2.1.3. Primer Design

In pHEE401E-UBQ_Bar vector cloning, the pCBC_DT1T2 vector was used as the insert template. The template was composed of the sgRNA, scaffold, terminator, promoter, and second sgRNA. In addition, the promoter was upstream of the vector’s multiple cloning site (MCS), and the scaffold and terminator were downstream of the vector’s MCS. When inserts amplified from pCBC_DT1T2 were cloned into the MCS of pHEE401E-UBQ_Bar, two sets of promoter–sgRNA–scaffold–terminators were automatically manufactured. An insert that starts with the sgRNA scaffold can be created by adjusting the annealing position of the forward primer. This scaffold-starting truncated insert can attach downstream of the original pHEE401E_UBQ_Bar insert, forming sequential promoter–sgRNA–scaffold–terminators sets (Figure 2). Theoretically, this combination can be extended continuously until it meets limiting factors, such as PCR length, vector size, and practical expression ratio of each sgRNA.

To design primers, the following general sequences (Table 1) are required.

The primers have a margin sequence to adjust the gap between the restriction-enzyme-recognition site and the cleavage site. The primer name ‘DT’ is a stand-in for the template plasmid, and the following number indicates which sgRNA is included in a primer. The sgRNA scaffold is abbreviated as ‘sc’. Forward and reverse primers are indicated with capital F and R, and the required restriction enzymes are denoted as Bs for *Bsa*I, Bb for *Bbs*I, and Aa for *Aar*I.

### 2.2. pBAtC_4sgRNA, pBAtC_6sgRNA Cloning

#### 2.2.1. Materials

pBAtC_tRNA (Appendix A) (this vector was kindly provided by Dr. Sang-Gyu Kim, Center of Genome Engineering, IBS, Daejeon, Korea).pTV00 (Appendix A) (this vector was kindly provided by Dr. Sang-Gyu Kim, Center of Genome Engineering, IBS, Daejeon, Korea).KOD-plus- (TOYOBO, Japan; Cat. No. KOD-201 200U).The QIAquick PCR Purification Kit (Cat. No. 28104).T4 DNA ligase (TaKaRa, Japan; Cat. No. 2011A).10X T4 DNA ligase buffer (TaKaRa, Japan; Cat. No. 2011A).*Aar*I restriction enzyme (Thermo Fisher, MA, USA; Cat. No. ER1581).*Bbs*I restriction enzyme (NEB, MA, USA; Cat. No. R0539S).

#### 2.2.2. Equipment

TaKaRa PCR Thermal Cycler Dice^®^ Touch (TaKaRa, Japan; Cat. No. TP350).Mupid-exU (Advanced, Japan; Cat. No. AD140).

#### 2.2.3. Primer Design

Unlike pHEE401E_UBQ_Bar primers, which have no distinct overhang sequence, pBAtC primers can create four different base-pair overhangs by dividing sgRNA into two. The dividing point should be in the middle four base pairs, but the region of the four base pairs can move laterally to avoid generating the same overhang sequence between the sgRNAs. These primers must include a type IIS restriction enzyme site that digests the DNA double-strand to expose a single strand and the desired four base-pair overhangs. One insert fragment comprised half of the upstream sgRNA, sgRNA scaffold, tRNA, and half of the downstream sgRNA in sequence (Figure 3).

The first insert fragment must have most upstream sgRNA intact, and the last insert fragment must have most downstream sgRNA intact. Owing to the distinctive sticky ends and non-reversible reaction features of type IIS restriction enzymes, the Golden Gate cloning of multiple inserts is possible for the simultaneous assembly of the insert polymer. In addition, to preserve the primer binding space on the very first and last sequence for full contact between the template and primer, the restriction-enzyme-recognition sites located in both terminal inserts should be different from those located between inserts. In this study, *Bbs*I and *Aar*I were used as the terminal restriction enzyme and inter-insert restriction enzyme, respectively. To design the primers, the following general sequences are required (Table 2 and Table 3).

In the primer name, ‘Bb’ and ‘Aa’ stand for the restriction-enzyme-recognition sequence for *Bbs*I and *Aar*I, respectively, in the primer. The abbreviations ‘g1g2’ or ‘g2g3’ indicate the number of sgRNAs included in the primer. For example, as a primer set that generates a product with sgRNA1 and sgRNA2, the primer has ‘g1g2’ in the middle of its name. The very first and last inserts require a second PCR to attach the *Bbs*I recognition site because of the long sequence of primers. Numbers 1 and 2 indicate the first and second PCRs, respectively.

For pBAtC_6sg vector construction, the primer sets ‘g1g2’ and ‘g2g3’ are identical to the pBAtC_4sg primer. The ‘g3g4’ primer set, formally designated as the terminal insert fragment in pBAtC_4sg, is modified as a median insert fragment. The primer naming abbreviations are the same as those for pBAtC_4sg.

## 3. Procedure

### 3.1. pHEE401E_4sg

1.Perform PCR with the following conditions (Table 4).

2.Check the band size (approximately 600 bp) by electrophoresis. If a solid band is presented, purify each PCR product using a PCR purification kit. Use the minimum elution amount for the maximum concentration.3.Ligate Insert 1 and Insert 2 using the Golden Gate reaction under the following conditions (Table 5).

4.Check the size of the ligation product using electrophoresis by loading 1 μL. The result may be multiple bands owing to unintended ligation and leftovers from the first inserts; however, if the 1200 bp size is the most dominant and presents as a solid band, proceed to the next step.5.Ligate the first product with the third insert under the following conditions (Table 6).

6.Perform gel extraction for the final Golden Gate product. The result may have three multiple bands and smearing, but if the 1800 bp size presents a band, proceed with gel extraction for this 1800 bp band. The final elution amount should be adjusted considering the DNA mass required in the next step.7.Ligate the insert polymer and the pHEE401E vector under the following conditions (Table 7).

8.Transform *Escherichia coli* with the ligated vector and spread on a kanamycin LB plate.9.Inoculate the well-grown colonies on new kanamycin LB plates and spectinomycin LB plates to screen the self-ligated vector transformants. Colonies that survive in kanamycin and die in spectinomycin are the colonies of interest (Figure 4).10.Proceed with colony PCR under the following conditions (Table 8).

11.Check the PCR product using gel electrophoresis. The result may show four multiple bands, but if the 2.5 kb size shows the most dominant band compared to the others, proceed with plasmid extraction for DNA sequencing.12.Confirm that all sgRNA sequences are presented in the DNA-sequencing result.

### 3.2. pBAtC_4sg, 6sg

1.Proceed with PCR under the following conditions (Table 9).

2.Proceed with Golden Gate reaction under the following conditions (Table 10).

3.Check the Golden Gate product using gel electrophoresis. The result may show multiple bands due to unintended ligation or residual inserts (one insert size is 200 bp), but if there is a 600 bp band in pBAtC_4sg or a 1000 bp band in pBAtC_6sg, proceed to the next step.4.Dilute the Golden Gate product 10 times as the Golden Gate template (GG template) and perform PCR under the following conditions (Table 11).

5.Digest the PCR product directly under the following conditions (Table 12).

6.Purify the *Bbs*I-digested insert polymer. Use the minimum elution amount for the maximum concentration.7.Ligate the purified insert polymer with the pBAtC_tRNA vector under the following conditions (Table 13).

8.Transform *E. coli* using the heat-shock method. Spread the transformed bacteria on spectinomycin LB plates.9.Check the transformant colony by colony PCR under the following conditions (Table 14).

10.Check the PCR product using gel electrophoresis. The result may show smear or multiple bands, but if the 1.1 kb band for pBAtC_4sg and the 1.5 kb band for pBAtC_6sg are the most dominant bands compared to the others, proceed with plasmid extraction for DNA sequencing.11.Confirm that all sgRNA sequences are present in the DNA sequencing result.

## 4. Expected Results

If all steps are successful, the DNA sequencing results should contain all designated sgRNA sequences (Appendix A).

## 5. Case Study of Multiple sgRNA Applications in Plants; Editing the Multigene of *FAD2* and *FATB* in Soybean

In this case, study, our aim was to develop soybean (*Glycine max)* due to its importance in biodiesel and human health by increasing the oleic acid and decreasing the saturated fatty acid content. To achieve this, fatty acid desaturase 2 (*FAD2*) and fatty acyl-ACP thioesterase B (*FATB*), which exist as multicopy genes in the soybean genome, need to be knocked out. FAD2 is an enzyme that catalyzes 18:1 to 18:2 fatty acids in the endoplasmic reticulum [9]. FATB is a plastid-located enzyme that has thioesterase activity to detach acyl carrier protein (ACP) and convert it to 16:0 and 18:0 free fatty acids from 16:0-ACP and 18:0-ACP [10,11]. The knockout of *FAD2* can block linoleic acid (18:2) synthesis and cause the accumulation of oleic acid (18:1). In addition, the knockout of *FATB* causes the deactivation of thioesterase and prevents the accumulation of saturated fatty acids in soybean seeds (Figure 5).

Before the design of sgRNAs, homologous *FAD2* and *FATB* genes were investigated due to the multicopy features of soybean. Using a database search and BLAST, a total of seven *FAD2* and four *FATB* were confirmed. In addition, gene expression data were examined for the determination of major and minor expression genes among multiple *FAD2* and *FATB* genes to knockout and remove pseudo-genes in silico. After filtering false data, five *FAD2* and four *FATB* were left, and sgRNAs were designed using RGEN Tools Cas-Designer [12]. Six sgRNAs were designed for multiple *FAD2* and *FATB* targets. sgRNA1 and sgRNA2 target *FAD2* and *FAD2-1*, while sgRNA5 targets the rest of the *FAD2* homologous genes. sgRNA3 and sgRNA4 target *FATB_a* and *FAT_b*, while sgRNA6 targets the remaining *FATB* homologous genes (Figure 6). According to this protocol, four sgRNAs corresponding to sgRNA1-4 were cloned into pHEE401E and pBAtC-tRNA to prepare pHEE401E_4sg and pBAtC_4sg, respectively. Then, six sgRNAs corresponding to sgRNA1-6 were cloned into the pBAtC-tRNA vector to construct pBAtC_6sg.

### 5.1. Construction of pHEE401E_4g

The very first PCR for each insert fragment was performed with the following primers (Table 15).

After the PCR with the conditions listed in Table 4, a 3 μL PCR product was examined by gel electrophoresis (Figure 7A). The PCR band was thick and clear at the desired size (~600 bp), allowing stepwise ligation with type IIS restriction enzymes.

As mentioned in the protocol section, the first and second fragments were ligated with the *Bbs*I*-*enzyme-mediated Golden Gate reaction. After the first ligation, a second ligation was performed with the *Aar*I enzyme-mediated Golden Gate reaction. A 3 μL sample of each reaction product was examined by gel electrophoresis (Figure 7B). Based on the band size of the Golden Gate reaction, each step showed the expected length (approximately 1.2 kb and 1.8 kp), leading to the next step.

The band with the proper size was extracted from the gel. The pHEE401E vector cloning, *E. coli* transformation, and selection were performed as described in the formal protocol chapter (pHEE401E vector procedure, steps 7–8). Colony PCR was performed using the conditions listed in Table 7, and the products were examined by gel electrophoresis (Figure 8C). Despite the multiple bands, expected bands (approximately 2.4 kb) were also shown to be relatively dominant compared to other bands. From these putative positive colonies, cloned pHEE401E_4sg vectors were extracted and sequenced for final verification. Sequencing also showed that four sgRNA sequences were successfully integrated.

After the construction ofpHEE401E_4sg was completed, it was transferred to *Glycine max* by Agrobacterium and callus co-culture [13].

### 5.2. Construction of pBAtC_4sg and pBAtC_6sg

The very first PCR for each insert fragment was performed with the following primers (Table 16 and Table 17).

After the PCR with the conditions listed in Table 8, a 3 μL PCR product was examined by gel electrophoresis (Figure 8A,B). The PCR band was thick and clear at the desired size (~200 bp), allowing for simultaneous Golden Gate ligation with type IIS restriction enzymes.

All insert fragments were ligated simultaneously with an *AarI-mediated* Golden Gate reaction (Table 9), and 3 μL of ligation products was examined by gel electrophoresis as described in the formal protocol (pBAtC_4sg, 6sg procedure, steps 2–3) (Figure 8C).

After the verification of the expected band size of the pBAtC_4sg insert and pBAtC_6sg insert (~550 bp and ~1 kb, respectively), PCR was performed under the conditions listed in Table 10. Finally, 3 μL of PCR products was examined using gel electrophoresis (Figure 8D).

Despite the multiple and smear bands presented, each insert size shows relatively thick and bright bands, allowing for the treatment of the *Bbs*I-restriction enzyme treatment and PCR purification (pBAtC_4sg, 6sg construction procedure, steps 5–7). After purification, each insert was cloned into the pBAtC_tRNA vector using the Golden Gate reaction and transferred to *E. coli*. Screening of positive colonies was performed with antibiotic LB agar plate and colony PCR (pBAtC_4sg, 6sg construction procedure, steps 8–10) (Figure 8E).

Despite the multiple bands, the expected bands (approximately 1.1 kb for pBAtC_4sg, 1.5 kb for pBAtC_6sg) were also relatively dominant compared to the other bands. From these putative positive colonies, cloned pBAtC_4sg and pBAtC_6sg vectors were extracted and sequenced for final verification. Sequencing also showed that four and six sgRNA sequences were successfully integrated. The construction of pBAtC_4sg and pBAtC_6sg was completed, and they were transferred to *Glycine max* by *Agrobacterium* and callus co-culture [13].

### 5.3. Deep Sequencing Analysis of FAD2 and FATB Multiple Target Genes

A total of 16 individuals were sampled from pHEE401E_4sg transgenic plants. Additionally, 27 and 29 individuals were sampled and examined from pBAtC_4sg and pBAtC_6sg transgenic plants, respectively. With sampled transgenic T0 generation soybean leaf, Genomic DNA (gDNA) was extracted using the CTAB solution [14]. Extracted gDNA was used as a template for the library construction for each sgRNA target site. The deep-sequencing platform was Illumina MiSeq, which uses the pair-end method. Pair-end FASTQ files were merged using the FASTQ-JOIN program (https://github.com/ExpressionAnalysis/ea-utils, accessed on 21 November 2021), and the indel ratio of each sgRNA target site was examined by CRIS.py [15].

The sgRNA target site for each gene is shown in Figure 7, and the indel ratios of each gene in the T0 transgenic plants of pHEE401E_4sg, pBAtC_4sg, and pBAtC_6sg are shown in Appendix A, respectively. Table 17 summarizes the indel ratios at the two positions of the target genes for pHEE401E_4sg-10, pBAtC_4sg-1, and pBAtC_6sg-5, which are representative lines of transformants for each vector. For each transformant, the indel ratio varied from as high as 100% to as low as 0.1%, depending on the target site (Appendix A). The average indel efficiency for all target genes for all the individuals of the T0 generation transformed by the three vectors was 44.2% for pHEE401E_4sg, 26% for pBAtC_4sg, and 22% for pBAtC_6sg. This result suggests that pHEE401E_4sg has the highest indel effect. There was no significant difference in the efficiency of sgRNA against multiple target genes, except for the *FATB*_*a* gene. This site was targeted by sgRNA 4 with a perfect base pair match but showed a lower indel ratio than the *FATB_b* second target site, which has one base pair mismatch with sgRNA4. The exact reason for this is still unknown, but we can assume that the *FATB_a* second target site is difficult to approach with sgRNA 4.

To verify the indel position and indel type percentage of each vector transformant, the FASTQ result files were investigated using the CRISPResso2 web tool [16]. However, the tool considers homologous genes such as *FAD2* and *FAD2-1* as base pair replacement mutations; therefore, the total indel ratio of each target site shows a different value from the CRISP.py (Table 18, Appendix A). That is, CRISPResso2 only counts perfect matching reference sequences as indels. For example, in the case of the pHEE401E_4sg-10 transformant, the indel ratio in the *FAD2* first target site using CRIS.py has a higher indel ratio value of 66.4% (Table 18), while the indel ratio with CRISPResso2 has a lower value of 23.12% (Figure 9). In addition, through the CRISPResso2 analysis, 29.13% of large deletions between the sgRNA1 and sgRNA2 target sites were confirmed at the second target sites of *FAD2* and *FAD2-1* (Figure 9). In this study, various indels in the T0 generation were identified. Such complex indels are expected to be fixed with one major indel per individual line as the generation progresses.

## 6. Conclusions

In this study, four to six sgRNAs were designed to knock out multiple soybean *FAD2* and *FATB* genes. Multiple sgRNAs were cloned into two different types of vectors, pHEE401E_UBQ_Bar and pBAtC_tRNA, using either stepwise or simultaneous Golden Gate ligation methods. Deep-sequencing analysis of the T0 generation soybean transformants of these three vectors shows that multiple genes were edited. These results suggest that this protocol can be useful for producing multiple sgRNAs for editing multiple genes in plants.

## Figures and Tables

**Figure 1 plants-10-02542-f001:**
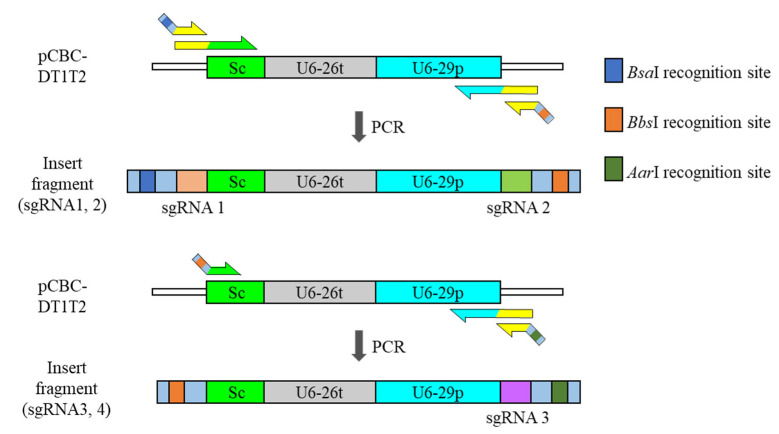
Schematic structure of each insert for pHEE401E ligation. Addition of fourth sgRNA is made possible by the switch restriction-enzyme-recognition site of *Bbs*I and *Aar*I. Sc; sgRNA scaffold, U6-26t; U6-26 terminator, U6-29p; U6-29 promoter.

**Figure 2 plants-10-02542-f002:**
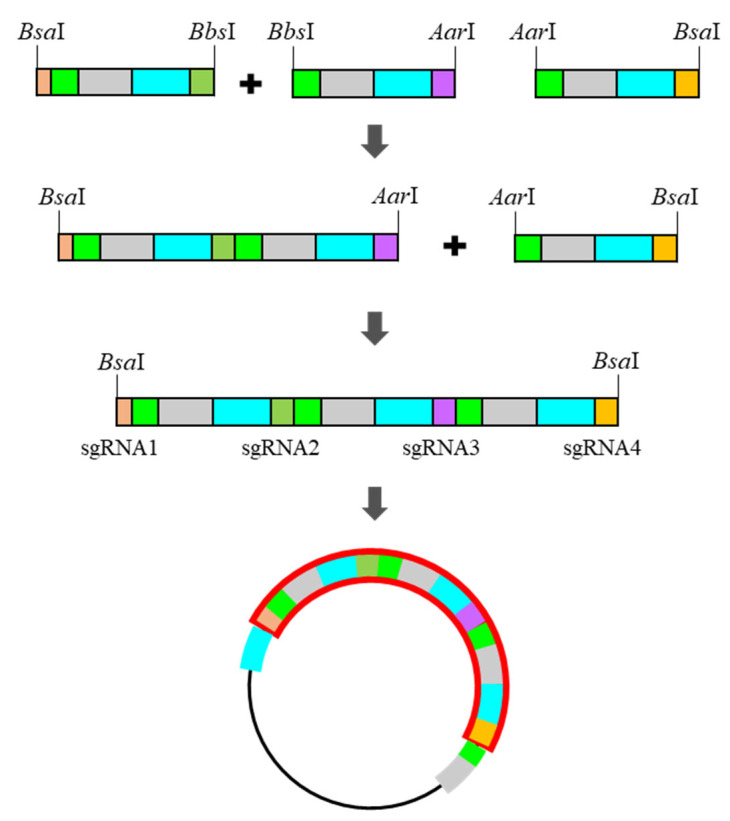
Stepwise ligation for ordered insert-polymer and final product with the pHEE401E_UBQ_Bar vector.

**Figure 3 plants-10-02542-f003:**
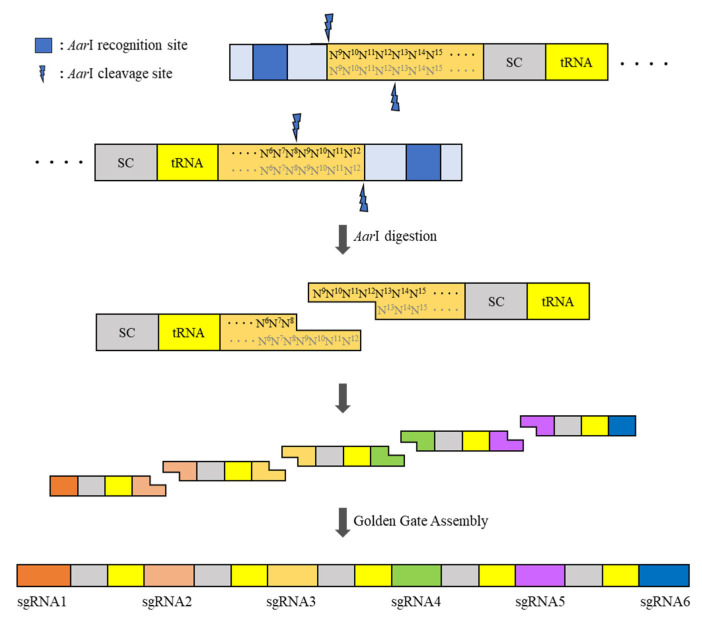
Schematic strategy of the construction of the pBAtC_tRNA vector-based multi-sgRNA insert.

**Figure 4 plants-10-02542-f004:**
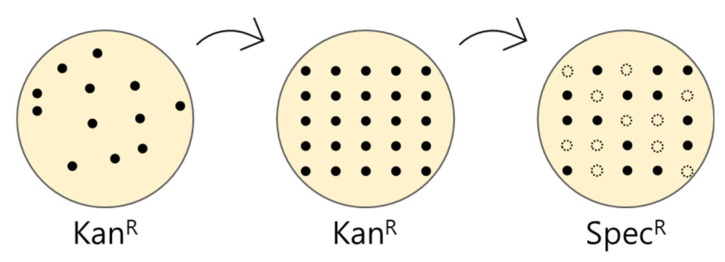
Schematic of the screening of the self-ligated vector transformant.

**Figure 5 plants-10-02542-f005:**
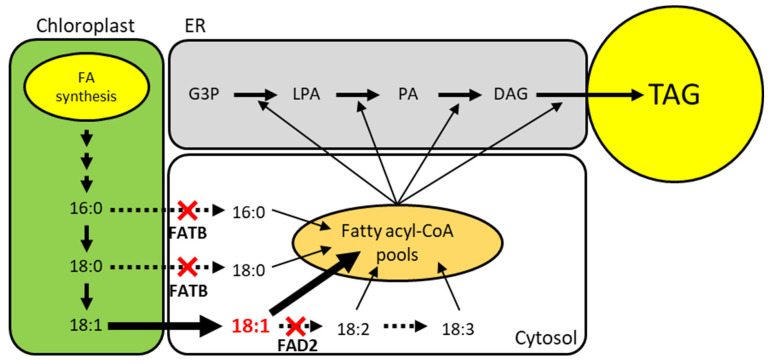
Schematic pathway of plant TAG biosynthesis and strategy to increase oleic acid and decrease saturated fatty acids. FAD2 and FATB knockout are indicated by red cross. Expected flux of fatty acid is depicted with arrows. The amount of flux is indicated by the thickness or type of the arrow. G3P; Glycerol-3-Phosphate, LPA; Lysophosphatidic Acid, PA; Phosphatidic Acid, DAG; Diacylglycerol, TAG; Triacylglycerol.

**Figure 6 plants-10-02542-f006:**
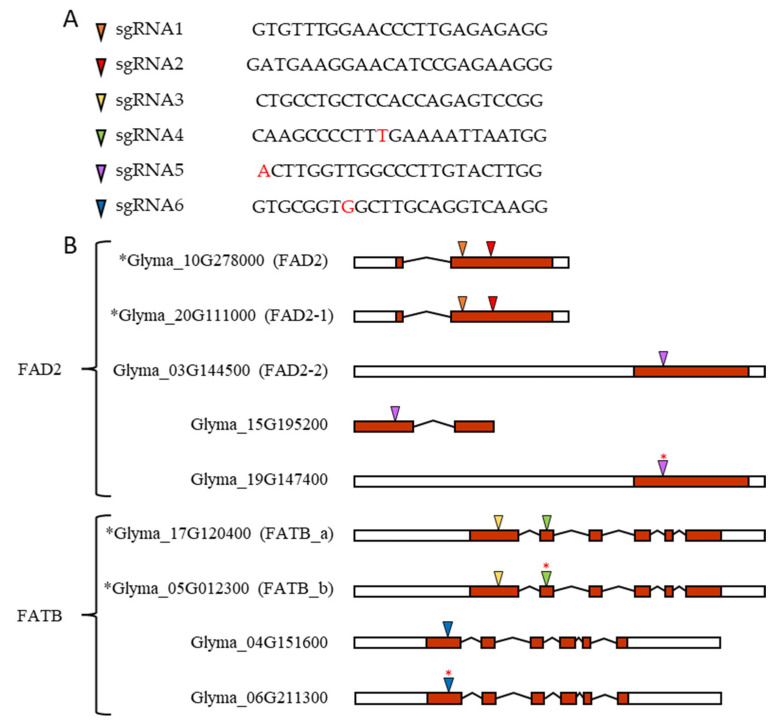
*FAD2* and *FATB* candidate genes in soybean and its sgRNA sequence and target location. (**A**) sgRNAs and their sequence. (**B**) Based on transcript form, left white box for 5′ end UTR, right white box for 3′ end UTR, red box for exon, angled line for intron splicing, and colored triangle for sgRNA target sites. Black asterisk in front of gene accession number indicates major expressing genes. Red asterisk on colored triangle indicates one base pair mismatch between gDNA sequence and sgRNA sequence. This mismatch is also indicated in red letters in the sgRNA4, sgRNA5, and sgRNA6 sequences in Figure 7A.

**Figure 7 plants-10-02542-f007:**
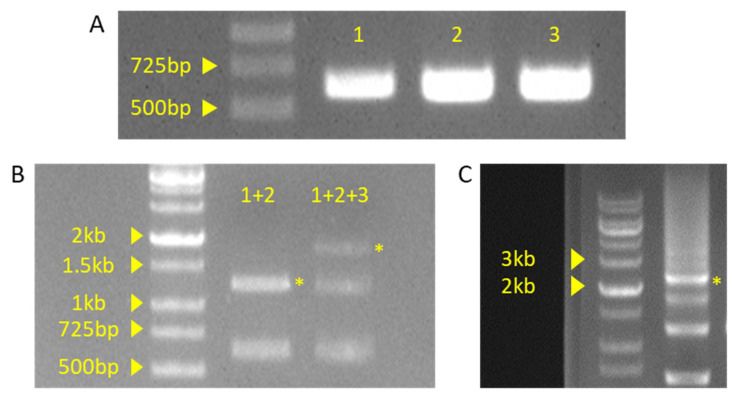
Gel picture of PCR or ligation products. (**A**) PCR product of insert fragment. Each fragment numbers indicate sequential order. 1: DT1 and DT2 primer set used fragment, 2: DT3 primer set used fragment, 3: DT4 primer set used fragment. (**B**) Product of Golden Gate reaction. +indicate ligation of insert fragment. (**C**) Product of pHEE401E vector colony PCR. Yellow asterisk indicated by the expected band is the expected result.

**Figure 8 plants-10-02542-f008:**
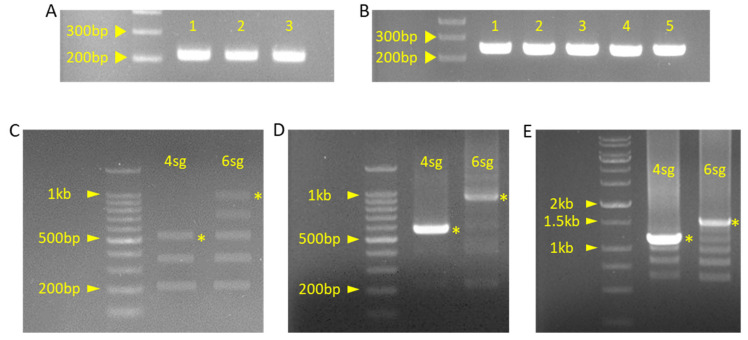
Gel picture of PCR or ligation products. (**A**) Product of insert fragment PCR for pBAtC_4sg construction. Each fragment number indicates a sequential order. 1: ‘g1g2’ primer set used fragment, 2: ‘g2g3’ primer set used fragment, 3: ‘g3g4’ primer set used fragment. (**B**) Product of insert fragment PCR for pBAtC_6sg construction. Each fragment number indicates sequential order. 1: ‘g1g2’ primer set used fragment, 2: ‘g2g3’ primer set used fragment, 3: ‘g3g4’ primer set used fragment, 4: ‘g4g5’ primer set used fragment, 5: ‘g5g6’ primer set used fragment. (**C**) Product of ligated insert fragment by Golden Gate reaction. (**D**) Product of ligated insert PCR of pBAtC_4sg and pBAtC_6sg. (**E**) Product of pBAtC_4sg, 6sg colony PCR. Yellow asterisk indicated by the expected band is the expected result.

**Figure 9 plants-10-02542-f009:**
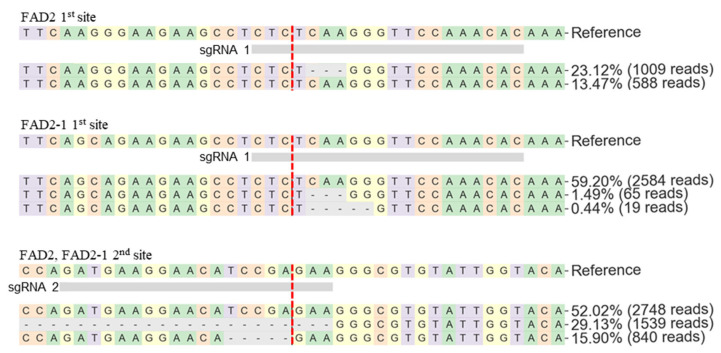
Target sequence alignment of pHEE401E_4sg-10 *FAD2* and *FAD2-1*. Vertical red dash indicates predicted cleavage site.

**Table 1 plants-10-02542-t001:** List of primers and sequences for the pHEE401E_UBQ_Bar vector.

Insert Name	Sequence
DT1-BsF	ATATATGGTCTCGATT(--sg1--)GTT
DT1-F	T(--sg1--)GTTTTAGAGCTAGAAATAGC
DT2-R	AAC(--sg2--)AATCTCTTAGTCGACTCTAC
DT2-BbR	ATTATTGAAGACNNAAAC(--sg2--)AA
DTsc_BbF	ATATATGAAGACNNGTTTTAGAGCTAGAAATAGCAAGTTAA
DT3-R	AAC(--sg3--)AATCTCTTAGTCGACTCTAC
DT3-AaR	ATTATTCACCTGCNNNNAAAC(--sg3--)AA
DTsc-AaF	ATATATCACCTGCNNNNGTTTTAGAGCTAGAAATAGCAAGTTAA
DT4-R	AAC(--sg4--)AATCTCTTAGTCGACTCTAC
DT4-BsR	ATTATTGGTCTCGAAAC(--sg4--)AA

Underlined sequences are restriction-enzyme-recognition sites.

**Table 2 plants-10-02542-t002:** List of primers and sequences for the pBAtC_4sg insert.

Insert Name	Sequence
Bb_g1g2_F1	(--sg1--)GTTTTAGAGCTAGAA
Bb_g1g2_F2	ATGAAGACNNTGCA(--sg1--)
Aa_g1g2_R	ATCACCTGCNNNN(--half of sg2, 5′ moiety--)TGCACCAGCCGGGAA
Aa_g2g3_F	ATCACCTGCNNNN(--half of sg2, 3′ moiety--)GTTTTAGAGCTAGAAATAGC
Aa_g2g3_R	ATCACCTGCNNNN(--half of sg3, 5′ moiety--)TGCACCAGCCGGGAATCGAA
Aa_g3g4_F	ATCACCTGCNNNN(--half of sg3, 3′ moiety--)GTTTTAGAGCTAGAA
Bb_g3g4_R1	(--sg4--)TGCACCAGCCGGGAATCGAA
Bb_g3g4_R2	ATGAAGACNNAAAC(--sg4--)

Underlined sequences are restriction-enzyme-recognition sites.

**Table 3 plants-10-02542-t003:** List of primers and sequences for the pBAtC_6sg insert.

Insert Name	Sequence
Aa_g3g4_F	ATCACCTGCNNNN(--half of sg3, 5′ moiety--)GTTTTAGAGCTAGAA
Aa_g3g4_R	ATCACCTGCNNNN(--half of sg4, 5′ moiety--)TGCACCAGCCGGGAATCGAA
Aa_g4g5_F	ATCACCTGCNNNN(--half of sg4, 5′ moiety--)GTTTTAGAGCTAGAA
Aa_g4g5_R	ATCACCTGCNNNN(--half of sg5, 5′ moiety--)TGCACCAGCCGGGAATCGAA
Aa_g5g6_F	ATCACCTGCNNNN(--half of sg2, 5′ moiety--)GTTTTAGAGCTAGAA
Bb_g5g6_R1	(--sg6--)TGCACCAGCCGGGAA
Bb_g5g6_R2	ATGAAGACNNAAAC(--sg6--)

Underlined sequences are restriction-enzyme-recognition sites.

**Table 4 plants-10-02542-t004:** PCR components and thermal conditions for the first insert.

Item	Concentration	Amount	Temperature	Time	Cycle
pCBC_DT1T2		(5 ng)	94 °C	2 min	1
DT1-BsF1 *	20 μM	1 μL	94 °C	15 s	30
DT1-F1 *	1 μM	1 μL	60 °C	20 s	
DT2-R1 *	1 μM	1 μL	68 °C	20 s	
DT2-BbR1 *	20 μM	1 μL	68 °C	2.5 min	1
10X KOD Buffer		5 μL	4 °C	Hold	1
dNTP	2 mM	4 μL			
MgSO4	25 mM	3 μL			
KOD-Plus-	1 U/μL	1 μL			
Distilled Water		to 50 μL			

* Primers must be substituted for the second and third insert fragments. In this case, the concentrations of DTsc-BbR and DTsc-AaR are 21 μM.

**Table 5 plants-10-02542-t005:** Golden Gate reaction components and thermal conditions.

Item	Concentration	Amount	Temperature	Time
Insert 1		(300 ng)	37 °C	5 h
Insert 2		(300 ng)	50 °C	5 min
10X T4 ligase Buffer		1.5 μL	80 °C	10 min
*Bbs*I	10 U/μL	1 μL	8 °C	Hold
T4 Ligase	350 U/μL	1 μL		
Distilled Water		to 15 μL		

**Table 6 plants-10-02542-t006:** Golden Gate reaction components and thermal conditions.

Item	Concentration	Amount	Temperature	Time
First golden gate product		7 μL	37 °C	5 h
Insert 3		(300 ng)	50 °C	5 min
50X oligonucleotide *	0.5 μM	0.5 μL	80 °C	10 min
10X T4 Ligase Buffer		2.5 μL	8 °C	Hold
*AarI*	2 U/μL	1 μL		
T4 Ligase	350 U/μL	1 μL		
Distilled Water		to 25 μL		

* 50X *Aar*I oligo must be added owing to its efficiency.

**Table 7 plants-10-02542-t007:** Golden Gate reaction components and thermal conditions.

Item	Concentration	Amount	Temperature	Time
Insert polymer		(200 ng)	37 °C	5 h
pHEE401E vector		(100 ng)	50 °C	5 min
10X T4 Ligase Buffer		1.5 μL	80 °C	10 min
*BasI*	20 U/μL	1 μL	8 °C	Hold
T4 Ligase	350 U/μL	1 μL		
Distilled Water		to 15 μL		

**Table 8 plants-10-02542-t008:** PCR components and thermal conditions for the first insert.

Item	Concentration	Amount	Temperature	Time	Cycle
Colony		(5 ng)	95 °C	5 min	1
U6-26P_F *	10 μM	1 μL	95 °C	30 s	30
UBQ10_R **	10 μM	1 μL	55 °C	30 s	
10X Buffer		2 μL	72 °C	2.5 kb ***	
dNTP	2.5 mM	1.6 μL	72 °C	5 min	1
DNA polymerase	5 U/μL	0.1 μL	8 °C	Hold	1
Distilled Water		to 20 μL			

* U6-26P_F sequence: AACCTTCAAGAATTTGATTGAATA; ** UBQ10_R sequence: CGTGTATTGAGCGTTGTTTA; *** use different elongation times based on the DNA-polymerase performance.

**Table 9 plants-10-02542-t009:** PCR components and thermal conditions for the first insert.

Item	Concentration	Amount	Temperature	Time	Cycle
pTV00		(5 ng)	94 °C	2 min	1
Bb_g1g2_F1 *	10 μM	2 μL	94 °C	15 s	30
Bb_g1g2_F2 *	1 μM	1 μL	60 °C	20 s	
Aa_g1g2_R *	10 μM	2.5 μL	68 °C	20 s	
10X KOD Buffer		5 μL	68 °C	2.5 min	1
dNTP	2 mM	4 μL	4 °C	Hold	1
MgSO4	25 mM	3 μL			
KOD -Plus-	1 U/μL	1 μL			
Distilled Water		to 50 μL			

* Three-primer PCR for the terminal insert primer only. In the case of pBAtC_4sg, the g1g2 and g3g4 sets were the terminal insert primers. In the case of pBAtC_6sg, the g1g2 and g4g6 sets were the terminal insert primers. If there are two primers that have the same direction, the outer and inner primer amounts should be 2 μL and 1 μL, respectively. The amount of the opposite direction of the solo primer is 2.5 μL. The median insert primer (10 μM) should be 2 μL in both directions.

**Table 10 plants-10-02542-t010:** Components of Golden Gate reaction and thermal conditions.

Item	Concentration	Amount	Temperature	Time	Cycle
Inserts *		(100 ng)	37 °C	5 min	40
50X oligonucleotide **	0.5 μM	0.3 μL	20 °C	10 min	
10X T4 Ligase Buffer		1.5 μL	20 °C	1 h	1
*AarI*	2 U/μL	0.5 μL	65 °C	20 min	1
T4 Ligase	350 U/μL	1 μL	8 °C	Hold	1
Distilled Water		to 15 μL			

* Every insert. ** Must be added owing to efficiency.

**Table 11 plants-10-02542-t011:** Components of PCR and thermal conditions.

Item	Concentration	Amount	Temperature	Time	Cycle
GG template	1/10 diluted	1 μL	94 °C	2 min	1
Aa_g1g2_F2	10 μM	2 μL	94 °C	15 s	40
Reverse primer *	10 μM	2 μL	60 °C	20 s	
10X KOD Buffer		5 μL	68 °C	1 min/kb **	
dNTP	2 mM	4 μL	68 °C	5 min	1
MgSO4	25 mM	3 μL	4 °C	Hold	1
KOD -Plus-	1 U/μL	1 μL			
Distilled Water		to 50 μL			

* pBAtC_4sg reverse primer: Aa_g3g4_R2. * pBAtC_6sg reverse primer: Aa_g5g6_R2. ** 4sg time: 40 s; 6sg time: 1 min 10 s.

**Table 12 plants-10-02542-t012:** Components of insert polymer digestion and thermal conditions.

Item	Concentration	Amount	Temperature	Time
PCR product	- *	20–25 μL	37 °C	2 h
10X *BsbI* Buffer		5 μL	65 °C	20 min
*BbsI*	10 U/μL	1 μL	8 °C	Hold
Distilled Water		to 50 μL		

* PCR product has no fixed concentration due to direct treatment with the restriction enzyme.

**Table 13 plants-10-02542-t013:** Golden Gate reaction components and thermal conditions.

Item	Concentration	Amount	Temperature	Time	Cycle
Insert polymer		(200 ng)	37 °C	5 min	40
pBAtC_tRNA		(100 ng)	20 °C	10 min	
50X oligonucleotide	0.5 μM	0.3 μL	20 °C	1 h	1
10X T4 Ligase Buffer		1.5 μL	65 °C	10 min	1
*Aar*I	1 U/μL	1 μL	8 °C	Hold	1
T4 Ligase	350 U/μL	1 μL			
Distilled Water		to 15 μL			

**Table 14 plants-10-02542-t014:** PCR components and thermal conditions.

Item	Concentration	Amount	Temperature	Time	Cycle
Colony		(5 ng)	95 °C	5 min	1
U6-26P_F	10 μM	1 μL	95 °C	30 s	30
NosP_Rev *	10 μM	1 μL	55 °C	30 s	
10X Buffer		2 μL	72 °C	See footer **	
dNTP	2.5 mM	1.6 μL	72 °C	5 min	1
DNA polymerase	5 U/μL	0.1 μL	8 °C	Hold	1
Distilled Water		to 20 μL			

* NosP_Rev sequence: AAGTCGCCTAAGGTCACT. ** pBAtC_4sg, expected PCR band size: 1159 bp. ** pBAtC_6sg, expected PCR band size: 1513 bp. ** Use different elongation times based on the DNA-polymerase performance.

**Table 15 plants-10-02542-t015:** List of primers and sequences for the pHEE401E_4sg construction.

Insert Name	Sequence
DT1-BsF	ATATATGGTCTCGATTGTGTTTGGAACCCTTGAGAGGTT
DT1-F	TGTGTTTGGAACCCTTGAGAGGTTTTAGAGCTAGAAATAGC
DT2-R	AACTTCTCGGATGTTCCTTCATCAATCTCTTAGTCGACTCTAC
DT2-BbR	ATTATTGAAGACATAAACTTCTCGGATGTTCCTTCATCAA
DTsc_BbF	ATATATGAAGACATGTTTTAGAGCTAGAAATAGCAAGTTAA
DT3-R	AACGACTCTGGTGGAGCAGGCACAATCTCTTAGTCGACTCTAC
DT3-AaR	ATTATTCACCTGCATATAAACGACTCTGGTGGAGCAGGCACAA
DTsc-AaF	ATATATCACCTGCATATGTTTTAGAGCTAGAAATAGCAAGTTAA
DT4-R	AACTTAATTTTCAAAGGGGCTTCAATCTCTTAGTCGACTCTAC
DT4-BsR	ATTATTGGTCTCGAAACTTAATTTTCAAAGGGGCTTCAA

Underlined sequences are restriction-enzyme-recognition sites.

**Table 16 plants-10-02542-t016:** List of primers for pBAtC_4sg insert fragments.

Insert Name	Sequence
Bb_g1g2_F1	GTGTTTGGAACCCTTGAGAGGTTTTAGAGCTAGAA
Bb_g1g2_F2	ATGAAGACATTGCAGTGTTTGGAACCCTTGAGAG
Aa_g1g2_R	ATCACCTGCATATTGTTCCTTCATCTGCACCAGCCGGGAA
Aa_g2g3_F	ATCACCTGCATATAACATCCGAGAAGTTTTAGAGCTAGAAATAGC
Aa_g2g3_R	ATCACCTGCATATTGGAGCAGGCAGTGCACCAGCCGGGAATCGAA
Aa_g3g4_F	ATCACCTGCATATTCCACCAGAGTCGTTTTAGAGCTAGAA
Bb_g3g4_R1	TTAATTTTCAAAGGGGCTTGTGCACCAGCCGGGAATCGAA
Bb_g3g4_R2	ATGAAGACATAAACTTAATTTTCAAAGGGGCTTG

Sticky ends that were generated by type IIS restriction enzyme are indicated with red text. Underlined sequences are restriction-enzyme-recognition sites.

**Table 17 plants-10-02542-t017:** Additional list of primers for pBAtC_6sg insert fragments.

Insert Name	Sequence
Aa_g3g4_F **	ATCACCTGCATATTCCA*CCAGAGTCGTTTTAGAGCTAGAA
Aa_g3g4_R **	ATCACCTGCATATTCAA*AGGGGCTTGTGCACCAGCCGGGAATCGAA
Aa_g4g5_F	ATCACCTGCATATTTGA*AAATTAAGTTTTAGAGCTAGAA
Aa_g4g5_R	ATCACCTGCATATAGCC*ACCGCACTGCACCAGCCGGGAATCGAA
Aa_g5g6_F	ATCACCTGCATATGGCT*TGCAGGTCGTTTTAGAGCTAGAA
Bb_g5g6_R1	AGTACAAGGGCCAACCAAGTTGCACCAGCCGGGAA
Bb_g5g6_R2	ATGAAGACATAAACAGTACAAGGGCCAACCAAGT

* Sticky ends generated by type IIS restriction enzymes are indicated with red text. ** Primer used as an alternative to pBAtC_4sg ‘g3g4’ primer set.

**Table 18 plants-10-02542-t018:** Indel ratio example of each of the three vector T0 transformants.

Target Genes	*FAD2-2*	*15G195200*	*19G147400*	*FAD2*		*FAD2-1*		*FATB_a*		*FATB_b*		*04G151600*	*06G211300*
				1st Site	2nd Site	1st Site	2nd Site	1st Site	2nd Site	1st Site	2nd Site		
pHEE401E_4sg-10	-	-	-	66.4	91.9	5.4	0.4	74	5.6	71.4	44.6	-	-
pBAtC_4sg-1	-	-	-	54	76.6	67.4	77.1	55.7	0.9	53.8	76.5	-	-
pBAtC_6sg-5	56.5	25.7	80.1	71.8	69	45.6	60	23.5	0.2	20.3	3.1	41.4	59.1

- indicates no indels detected. Ratio unit: percent (%).

## Data Availability

The data presented in this study are available in the Appendix A.

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
