# Peer review of "Construction of Multiple Guide RNAs in CRISPR/Cas9 Vector Using Stepwise or Simultaneous Golden Gate Cloning: Case Study for Targeting the FAD2 and FATB Multigene in Soybean"

_plants, 2021, doi:10.3390/plants10112542_

Round 1
Reviewer 1 Report
The revised version still lacks novelty and the author's text update suggesting their protocol doesn't require sub-cloning is insufficient and unclear. Several labs have adopted golden gate strategies where one pot reaction can be performed to clone multiple fragments with appropriate use of restriction enzymes. Also please address all reviewer's comments point by point. Specifically addressing how the authors think the protocol is novel and helps the field of CRISPR application in plants.
Author Response
Comments and Suggestions for Authors
The revised version still lacks novelty and the author's text update suggesting their protocol doesn't require sub-cloning is insufficient and unclear. Several labs have adopted golden gate strategies where one pot reaction can be performed to clone multiple fragments with appropriate use of restriction enzymes. Also please address all reviewer's comments point by point. Specifically addressing how the authors think the protocol is novel and helps the field of CRISPR application in plants.
(Answer) This protocol does not require a subcloning vector for Golden Gate assembly because it directly connects multiple gRNAs to the vector with the existing other method for producing multiple gRNAs. Therefore, this protocol (1) no subcloning vector is required and (2) it is possible to manufacture with a general CRISPR/cas9 vector. (3) The advantage is that even beginners can follow this protocol.
This distinction is described in the last sentence of the Abstract, and is additionally described in the Introduction. Edited sentences are marked in red.
Reviewer 2 Report
The main issue is the presentation of the in planta case study. 1) Although Fig 7 displayed the diagram of the sgRNA target location, the detalied sequence information of each sgRNA is still lacking. At least, it is hard to find if the sgRNA sequence of FAD2 or FAD2-1 is identical or not. 2) As shown in Fig.7, FAD2 and FAD2-1 are targetd by sgRNA1 and sgRNA2, while the left three FAD2 like genes are targeted by sgRNA5, similarly, FATB_a and FATB_b are targted by sgRNA3 and sgRNA4, while the left two FATB genes are targetd by sgRNA6. But in the text, such a clear description is lacking. 3) Fig 8-13 are gel pictures describing each step of the constuction of the corresponding vectors, which could be emerged to maximum two figures, one relevant to construction of pHEE401E_4g, the other related to pBAtC_4sg pBAtC_4sg and pBAtC_6sg. 4) Regarding to seep sequencing analysis, sequence inforation of each targeted gene is missing. From the supplementary tables, readers can not judge if the knockout (indel) occur in place.
English of this MS needs significant improvement. For example, in the legend of Fig. 7, "The genes that designated simulatanously by two sgRNAs, has first and second site which start from 5' end"; line 326-327, "total six sgRNAs were designed for multiple FAD2 and FATB knockout".
Author Response
The main issue is the presentation of the in planta case study.
1) Although Fig 7 displayed the diagram of the sgRNA target location, the detalied sequence information of each sgRNA is still lacking. At least, it is hard to find if the sgRNA sequence of FAD2 or FAD2-1 is identical or not.
(Answer)We added sgRNAs sequence information and sgRNAs target position for each gene in Figure 7. And more information was added in the legend.
2) As shown in Fig.7, FAD2 and FAD2-1 are targetd by sgRNA1 and sgRNA2, while the left three FAD2 like genes are targeted by sgRNA5, similarly, FATB_a and FATB_b are targted by sgRNA3 and sgRNA4, while the left two FATB genes are targetd by sgRNA6. But in the text, such a clear description is lacking.
(Answer) Information for each sgRNA target is described in the revised manuscript. It is described at the bottom of page 12. We have rewritten the sentences that the reviewer pointed out as unclear.
3) Fig 8-13 are gel pictures describing each step of the construction of the corresponding vectors, which could be emerged to maximum two figures, one relevant to construction of pHEE401E_4g, the other related to pBAtC_4sg pBAtC_4sg and pBAtC_6sg.
(Answer) Figures are merged in one figure for one vector type.
4) Regarding to deep sequencing analysis, sequence inforation of each targeted gene is missing. From the supplementary tables, readers can not judge if the knockout (indel) occur in place.
(Answer) The trend of the indel sequence of the target site was added to Figure 10. Representation of multigene indel sequences in T0, a still segregated generation, may be incomplete. Please understand that the indel sequences to be analyzed are so vast that only the indel sequences of FAD2 and FAD2-1 of the pHEE401E_4sg-10 line are representatively analyzed and illustrated.
English of this MS needs significant improvement. For example, in the legend of Fig. 7, "The genes that designated simulatanously by two sgRNAs, has first and second site which start from 5' end"; line 326-327, "total six sgRNAs were designed for multiple FAD2 and FATB knockout".
(Answer) We have edited the content of the sentence you pointed out to make it easier to understand. And the English text was reviewed again for the entire manuscript.
Round 2
Reviewer 2 Report
Two concerns remain.
1) Page 2 line 21, does pBAtC-tRNA vector also have a type IIS restriction enzyme recognition site for ligation?. If yes, please write it out. If not, please revise the abovementioned sentence (Page 2 line 12-13) that using type IIS restriction enzyme.
2) Page 13 Fig 7A, what do these red colored letters in sgRNA4-6 mean?
In addtion, the English writing needs significant improvement. For example,
1) Page 1 Line 16, add “a” after “and”and before “polycistronic-tRNA-gRNA strategy”;
2) Page 2 line 5, change “start” to “starts’;
3) Page 2 line 12-13, rewrite the sentence “there is a need for a method for cloning multiple sgRNAs using type IIS restriction enzymes even in a laboratory that has only a general CRISPR/Cas9 vector” to emphasize the significance of this research;
4) Page 2 line 20, change “two vector” to “two vectors”;
5) Particularly in Page 17 line 5-15, significant improvement in English writing is needed. It’s hard to get the take-home message.
Author Response
Response to review
Reviewer 2
Thank you for the detailed review. The edited parts are marked in red in the manuscript.
Two concerns remain.
1) Page 2 line 21, does pBAtC-tRNA vector also have a type IIS restriction enzyme recognition site for ligation? If yes, please write it out. If not, please revise the abovementioned sentence (Page 2 line 12-13) that using type IIS restriction enzyme.
(Answer) The pBAtC-tRNA vector also has a type IIS AarI restriction enzyme site. The sentence on lines 21-23 of Page 2 has been rewritten.
2) Page 13 Fig 7A, what do these red colored letters in sgRNA4-6 mean?
(Answer) The nucleotide sequence shown in red in the gRNA sequences of sgRNA4-6 indicates a single nucleotide mismatch at the target position of the genes in Figure 7B. This is explained in the legend of Figure 7.
In addtion, the English writing needs significant improvement. For example,
(Answer) The entire manuscript has been reviewed and proofread by native English speakers to improve readability.
1) Page 1 Line 16, add “a” after “and”and before “polycistronic-tRNA-gRNA strategy”;
2) Page 2 line 5, change “start” to “starts’;
(Answer) We have corrected the English in the two cases you pointed out.
3) Page 2 line 12-13, rewrite the sentence “there is a need for a method for cloning multiple sgRNAs using type IIS restriction enzymes even in a laboratory that has only a general CRISPR/Cas9 vector” to emphasize the significance of this research;
(Answer) We have rewritten the sentence to emphasize the significance of our research.
4) Page 2 line 20, change “two vector” to “two vectors”;
(Answer) The sentence has been corrected as you pointed out.
5) Particularly in Page 17 line 5-15, significant improvement in English writing is needed. It’s hard to get the take-home message.
(Answer) We have rewritten and revised the contents of Page 17 line 35-44, and Page 18 line 1-2 as you pointed out.
This manuscript is a resubmission of an earlier submission. The following is a list of the peer review reports and author responses from that submission.
Round 1
Reviewer 1 Report
In this manuscript, the authors provide a cloning approach to construct a vector with multiple guide RNAs for targeting multiple genes. The protocol provided is well written and provides adequate details for reproducibility. However, the manuscript lacks novelty. Several papers have been published that describe ways to make multiplex guide RNA vectors. Few examples:
https://www.nature.com/articles/s41598-018-35727-3
https://plantmethods.biomedcentral.com/articles/10.1186/s13007-020-00580-x
https://www.ncbi.nlm.nih.gov/pmc/articles/PMC6141065/
It is unclear how the described approach is different or better than published approaches.
Reviewer 2 Report
This MS aimed to use two strategies to construct two different vectors using either one promoter per sgRNA or polycistronic tRNA-gRNA strategies, to simultaneously target multiple genes in plants by using multiple sgRNAs in a vector.
The main problem is that except description of the principle and the procedures, no any solid evidence to show that these stratagies work in vitro. Let alone its success in planta.
In addition, the presentation of this work needs to be improved. For example, in Figure 1, What is U6-29p?Where is U6-26p?
Color for each sgRNA should be different.
There are many grammar issues: for example in the Abstract, this sentence "Four sgRNAs containing the pHEE401E_UBQ_Bar vector and four to six sgR-15 NAs containing the pBAtC_tRNA vector were constructed" needs to be revised. In Introduction section, “which fundamentally reform or delete original base sequences” needs to be corrected.